# Modeling and Experiments on Temperature and Electrical Conductivity Characteristics in High-Temperature Heating of Carbide-Bonded Graphene Coating on Silicon

**DOI:** 10.3390/mi15060673

**Published:** 2024-05-22

**Authors:** Lihua Li, Ruiying Wang, Yingwei Huang, Xingbang Li

**Affiliations:** Sino-German College of Intelligent Manufacturing, Shenzhen Technology University, Shenzhen 518118, China; wangruiying1002@foxmail.com (R.W.);

**Keywords:** non-isothermal glass hot embossing, carbide-bonded graphene, electro-thermal conversion characteristics

## Abstract

A novel non-isothermal glass hot embossing system utilizes a silicon mold core coated with a three-dimensional carbide-bonded graphene (CBG) coating, which acts as a thin-film resistance heater. The temperature of the system significantly influences the electrical conductivity properties of silicon with a CBG coating. Through simulations and experiments, it has been established that the electrical conductivity of silicon with a CBG coating gradually increases at lower temperatures and rapidly rises as the temperature further increases. The CBG coating predominantly affects electrical conductivity until 400 °C, after which silicon becomes the dominant factor. Furthermore, the dimensions of CBG-coated silicon and the reduction of CBG coating also affect the rate and outcome of conductivity changes. These findings provide valuable insights for detecting CBG-coated silicon during the embossing process, improving efficiency, and predicting the mold core’s service life, thus enhancing the accuracy of optical lens production.

## 1. Introduction

A novel non-isothermal glass hot embossing system has been proposed that utilizes intrinsic silicon as a mold core. The silicon mold core is coated with a three-dimensional carbide-bonded graphene (CBG) coating using a chemical vapor deposition (CVD) method [1]. The choice of intrinsic silicon as the mold core is based on its high thermal conductivity, which facilitates efficient heat transfer during the embossing process, resulting in high-precision embossing. The CBG coating acts as a thin-film resistance heater, enabling rapid heating of the mold core above the glass sample’s transition temperature, even with a low applied supply voltage [2]. Additionally, Yao et al. [3] has shown that using CBG coating can significantly reduce the duration of the non-isothermal hot embossing process to approximately 3 s by optimizing preheating and cooling times. This novel system offers improved temperature control accuracy and process quality for glass products, reduced energy consumption, and shorter processing cycles compared to traditional approaches. However, several challenges need to be addressed. One challenge is the negative temperature characteristic of silicon’s electrical conductivity, which affects the performance of the CBG-coated silicon mold core system. The resistance of the system sharply decreases with increasing temperature, leading to temperature control instability when a constant power or voltage is applied. This can result in mold expansion and deformation and affect the accuracy of the glass product’s size and shape at high temperatures (700 °C). Additionally, the current flow path in the system changes dynamically during the heating process, causing an uneven distribution of the temperature and thermal stress fields. The high thermal stress levels can lead to the rupture of the brittle silicon mold cores and the formation of bubbles or breakage in the glass samples, negatively impacting the overall product quality. Further investigation is needed to understand the relationship between temperature and electrical conductivity of CBG-coated silicon in non-isothermal glass hot embossing systems to precisely control the current for a steady temperature rise. This research can help determine if the CBG coating undergoes any damage during the embossing process and predict the mold core’s service life for timely recoating. These insights can contribute to the development and application of CBG coating, enabling more efficient and reliable non-isothermal glass hot embossing systems.

Regarding the research on the interaction between CBG coating and intrinsic silicon, Huang et al. [4] conducted initial research in 2013. They found that CBG coating networks on silicon wafers exhibited extremely high electrical and thermal conductivity. For example, a nearly 45 nm thick CBG coating on a silicon wafer exhibited a measured electrical conductivity of 1.98 × 10^4^ S/m and a resistance of only 20.4 Ω. Xie et al. [5,6] used a novel CBG coating technology on micro-patterned silicon stampers. They observed that the temperature of the CBG coating increased with the applied voltage, and the heating rate significantly increased with increasing voltage. The heating rate could reach an average of 10 °C/s and an instantaneous rate of 16.1 °C/s as the silicon wafer heated from 60 °C to 140 °C. The surface electrical resistance of the CBG coating on a thick silicon wafer was much lower than on a thin one. Garman et al. [7] discovered the mechanism of SiOC for accelerating the formation of CBG coating and manipulating the transition layer and CBG coating properties through tailored feeding strategies and growth time of SiOC. They found that both silicon and quartz could be coated with a thicker, more thermally conductive, and better quality CBG coating. The heating rate of the CBG coating increased with the applied voltage, resulting in a higher maximum temperature over time [8]. Additionally, simulations [9] have been conducted to explore the correlation between the mold core’s surface temperature, current, resistance, and voltage under stable conditions. Previous studies have primarily focused on assessing heating rates and understanding the relationship between temperature, current, and voltage in CBG-coated silicon. However, there is a knowledge gap regarding how temperature and electrical conductivity interact between CBG coating and silicon under high-temperature heating conditions. Furthermore, the mechanisms for improving heat transfer efficiency at different temperatures have not been fully explored. Further research into the electrothermal conversion characteristics of CBG coatings and the development of advanced temperature control methods are needed.

In this study, we used finite element analysis (FEA) with COMSOL Multiphysics software 6.0 to develop a comprehensive numerical model of Joule heating-based heat transfer. Our focus was on understanding the relationship between temperature and electrical conductivity of CBG-coated intrinsic silicon at a high temperature of 700 °C. In our simulation model, we treated the CBG coating as a thin-layer structure and employed FEA techniques to explore the electro-thermal conversion characteristics of CBG coatings. To validate our simulation model, we conducted an experiment to assess the consistency between the simulation results and experimental results, as well as to identify any other factors that could affect the outcomes. Additionally, in the second experiment, our goal was to compare the electro-thermal conversion characteristics and size between silicon and CBG-coated silicon. By establishing a correlation between electrical conductivity and temperature, we aimed to enable high-precision closed-loop temperature control during the glass hot pressing process. In the third experiment, we compared the differences in electro-thermal conversion characteristics under normal and abnormal conditions. This was aimed at detecting potential damage and predicting the service life of the CBG-coated silicon mold core. The final outcomes would improve the efficiency and consistency of the non-isothermal hot embossing process.

## 2. Materials and Methods

### 2.1. Finite Element Simulation of Heating Process

To gain a comprehensive understanding of the heating process and the variation and relationship between temperature and electrical conductivity, a simulation model was developed. This model included quartz, intrinsic silicon, CBG coating, and copper electrodes, as shown in Figure 1a. By simulating the entire electric heating process, this model enables accurate predictions of temperature and electrical conductivity variations, allowing for informed decision-making and effective process control.

The CBG coating exhibits significantly higher conductivity than intrinsic silicon at room temperature [10]. Within the extrinsic region, where the surface temperature of the CBG coating ranges from 20 °C to approximately 60 °C, the carrier concentration remains nearly constant due to the energization of dopant carriers into the conduction band, resulting in minimal thermal generation of additional carriers. As a result, the resistance of the CBG-coated silicon exhibits a slightly fluctuating but constant behavior within this temperature range. However, upon application of an input current voltage and subsequent temperature increase, the extrinsic region transitions into the intrinsic region, supplying greater thermal energy to the intrinsic silicon and CBG coating. This leads to an exponential rise in the electrical conductivity of both the intrinsic silicon and the CBG coating, consequently generating a significant amount of Joule heat. To investigate the temperature-dependent electrical conductivity of a CBG coating, we deposited it on a quartz substrate with dimensions matching those of intrinsic silicon. By applying a current and voltage to both sides of the CBG coating (Figure 2), we heated it up to 700 °C and recorded its conductivity. Our experimental results showed that the CBG coating maintained a constant resistance of approximately 20 Ω throughout this process.

According to previous research [11], an equation for the electrical resistivity of single-crystal and polycrystalline silicon had been established. This equation can be used to derive the tendency of electrical conductivity change with temperature for silicon. Additionally, it is known that the conductivity of the CBG coating can be calculated by determining the resistivity and Hall mobility [12,13]. The resistivity of the graphene network is obtained by multiplying the sheet resistance by the thickness of the CBG coating.

The heating process of CBG-coated silicon was simulated using COMSOL. Although this simulation did not consider the covalent bonds between the CBG coating and the silicon substrate, it still reflected some significant characteristics. The impact of the mold holder was found to be negligible and was therefore ignored. Given that the thickness of the CBG coating was only 200 nm, it was treated as both a resistive and thermal thin layer. The Joule heat equations of both the CBG coating and silicon are expressed as follows:(1)$$\left\{ \begin{array}{l} C_{p}\frac{\partial T}{\partial t}=Q_{j}+S_{1}\frac{\partial }{\partial x}\cdot \left(\Lambda \frac{\partial T}{\partial x}\right)-\sigma_{SB}S_{2}\left(T_{1}^{4}-T_{0}^{4}\right)-\lambda S_{3}\left(T_{1}-T_{0}\right)\\ \nabla_{t}\cdot \left[d_{s}\left(\sigma_{s}E+\varepsilon_{0}\varepsilon_{r}\right)\frac{\partial E}{\partial t}+J\right]=0\\ Q_{j}=\frac{1}{2}Re\left(J\cdot E^{*}\right)\\ E=-\nabla_{t}V \end{array}\right.$$
where *c_p_*_,*s*_ = (*d_s_c_p_*_,*s*_*ρ_s_* + *d_g_c_p_*_,*g*_*ρ_g_*)/(*d_s_* + *d_g_*) and Λ=(*d_s_λ_s_* + *d_g_λ_g_*)/(*d_s_* + *d_g_*), *d_s_*, *c_p_*_,*s*_, *ρ_s_*, *λ_s_* are the thickness (m), density (kg·m^−3^), thermal conductivity (W·m^−1^·K^−1^), and the specific heat capacity (J·kg^−1^·K^−1^) of silicon; *d_g_*, *c_p_*_,*g*_, *ρ_g_*, *λ_g_* are the thickness (m), density (kg·m^−3^), thermal conductivity (W·m^−1^·K^−1^), and the specific heat capacity (J·kg^−1^·K^−1^) of CBG coating; *S*_1_ is the is the cross-sectional area of the silicon mold core (m^2^); *σ_SB_* is the Stefan–Boltzmann constant; *S*_2_ is the surface area of the silicon mold core (m^2^); *λ* is the heat transfer coefficient; *S*_3_ is the inner surface area of the chamber (m^2^); *T*_0_ is the air temperature in the chamber (K); *T*_1_ is the temperature of silicon mold core (K); *Q_j_* is the heat source of silicon mold core (W·m^−3^); ∇*_t_* is the tangential gradient operator; *σ_s_* is the conductivity of silicon mold core (S·m^−1^); *E* is the electric field intensity of silicon mold core (V·m^−1^); *ε*_0_ is the dielectric constant of silicon mold (F·m^−1^); *ε_r_* is the relative permittivity of silicon mold core; *J* is the current density of silicon mold. *V* is the electric potential of silicon mold core.

In the simulation, the resistivity of the intrinsic silicon substrate was assumed to be larger than 5000 Ω at room temperature, and the thickness of the CBG coating was 200 nm, consistent with the experimental conditions. The surface temperature data (referred to as *T*) obtained during the heating process in COMSOL, were exported to MATLAB R2022a for further analysis. Based on findings from prior research [14], it has been observed that when the temperature is higher than 400 K, the electrical conductivity of semiconductors exhibits a temperature-dependent variation, as described by the expression exp(−*E_g_*/(2·*k_B_*·*T*)). It is noted that intrinsic silicon follows this principle as well. Since intrinsic silicon and the CBG coating can be simply regarded as parallel resistances, the electrical conductivity is described by Equation (2). We used k_c_ as a coefficient to correct for dimensional differences and applied this equation to achieve curve fitting.
(2)$$\sigma =\frac{1}{R_{eq}}={R_{si}}^{-1}+{R_{CBG}}^{-1}={\left(k_{c}\exp(\frac{-E_{g}}{2k_{B}T})\right)}^{-1}+{R_{CBG}}^{-1}$$
where *R_si_* is determined inversely proportion to the Arrhenius Equation, while *R_CBG_* is measured as 20 Ω; *k_c_* is a constant, fitted to 349.3; *E_g_* represents the energy band gap of silicon, fitted to 0.7137 eV; *k_B_* is Boltzmann constant. Once the surface temperature (*T*) of the CBG-coated silicon from Equation (1) is obtained, we can calculate the electrical conductivity of the CBG-coated silicon.

### 2.2. Heating Process and Experiment Setup

Firstly, a 200 nm thick layer of CBG networks was coated on the surface of the 18 × 18 × 11 mm intrinsic silicon mold core by CVD. The thickness of the CBG coating was determined by the reaction time and CVD growth temperature. It was found that a 150 nm layer of CBG coating could be formed on the silicon surface by CVD at a temperature of 980 °C and a reaction time of 30 min. The detailed process of the CVD method used to prepare the CBG coating has been described by Huang et al. [4]. The experiment was conducted using a customized non-isothermal glass hot embossing machine, as depicted in Figure 3. The CBG-coated silicon mold core was placed in the vacuum chamber of a specially designed non-isothermal glass hot embossing machine, which maintained a vacuum pressure of approximately 1 MPa (Figure 1b). The braid copper strips were connected to the CBG coated silicon as electrodes. Subsequently, an electric current was passed through the copper electrodes to silicon mold core from both sides, with the CBG coating serving as the electrical conductor and generating heat in accordance with Joule’s law. To prevent the dissipation of heat and electricity, the high electrical and thermal resistance quartz glass was used to insulate the CBG coated silicon from the surrounding environment. Furthermore, the quartz glass beneath the CBG coated intrinsic silicon served as a support structure.

The two copper electrodes of the intrinsic silicon were connected to an external power supply with a maximum output voltage of 100 V and output current of 30 A through two high-temperature wires wrapped in fiberglass sleeves. This power supply was also connected to the 220 V power output, following the principle of P = U∙I. The value of the output power was dictated by the power curve, which was simplified to be proportional to the heating time. Specifically, the output power was expressed as P(t) = kt, where k was a proportional constant adjustable by the user. Notably, the shorter the heating cycle, the higher the value of constant k. However, if k was set to be too high, the sample might be burned, causing a potential threat to the experiment’s validity.

There were three different online infrared thermometers to measure the surface temperature, which can be seen in Figure 4. An online infrared thermometer was placed at the center of the coating surface to measure the surface temperature of the glass sample (T_1_). Another online infrared thermometer (T_2_) was positioned at the middle of the mold core to record the temperature of the CBG-coated silicon. In addition, the surface temperature of the mold holder was measured by temperature (T_3_). Following the cold junction compensation, the online infrared thermometer signals were collected using a multi-channel data acquisition device with a sampling interval of 200 ms. In addition to temperature measurements, the data logger also recorded the heater voltage, heater current, mold temperature, vacuum pressure, mold displacement, and press force. To ensure the validity of the data, the entire heating and cooling process was performed multiple times.

## 3. Results and Discussion

The objective of this study is to investigate the electro-thermal conversion characteristics of CBG-coated silicon. To explore the underlying rules, two comparative experiments were conducted. The aim of the first experiment was to evaluate the correlation between experimental findings and simulation results and confirm that external factors do not influence the trend of electrical conductivity in CBG-coated silicon with respect to temperature. The second experiment aimed to identify the differences of the electro-thermal conversion characteristics and size between silicon and CBG-coated silicon.

Through a thorough analysis of FEA models, we observed that the surface temperature of the silicon with CBG coating increased to nearly 700 °C in just 90 s, as depicted in Figure 5. Thereafter, temperature and electrical conductivity data that were calculated in COMSOL were exported to MATLAB for analysis. By combining the temperature and electrical conductivity data, we obtained a graph illustrating the electrical conductivity changes with temperature, as shown in Figure 6a. The graph demonstrated that the calculated and experimental data were in good agreement; the electrical conductivity increased with the temperature steadily. When the temperature exceeded 300 °C, the rate of increase in electrical conductivity accelerated. According to the above description of the electrical and thermal responses, the heating behavior of the CBG coated silicon resembled that of semiconductor materials. At room temperature, its electrical conductivity falls between that of a conductor and an insulator, and it increases rapidly with rising surface temperature. In addition, its excitation temperature was relatively low, less than 150 °C, revealing the small energy band gap of the CBG-coated silicon.

The consistency of electrical conductivity with temperature in both simulated and experimental results led us to further verify that the electrothermal properties of the CBG-coated silicon are responsible for this trend. Hence, we conducted experiments comparing the changes between temperature and electrical conductivity with and without an upper mold and examined how the presence of an upper mold at distances affected electrical conductivity with temperature. The detailed experimental setup is in Table 1, and further illustrated in Figure 7. The initial position was set to 0 by aligning the clamping positions of both the upper and lower molds. The experimental model was consistent with the simulated model in COMSOL, and the heating process was identical. After importing experimental data from Table 1 into MATLAB for calculation and graphing, we observed a consistent trend in the heating process between our calculations and experiments. As shown in Figure 6b, the trend of the two curves (group 1 to group 4) was similar, with electrical conductivity gradually increasing with temperature. This suggests that the position of the lower mold and the heating temperature do not influence the change of conductivity with temperature. In addition, the presence of an upper mold did not affect the accuracy of the temperature-electrical conductivity curve (group 5 and group 6). In another set of comparative experiments, we varied the distance between the upper and lower molds and the temperature of the lower and the upper, for a total of four experiments (group 7, group 8, group 9 and group 10). We found that the different positions and temperatures of the mold did not affect the trend of the curve. In conclusion, all curves of electrical conductivity changes with temperature from different conditions were the same. Our experimental data confirmed the accuracy and reliability of our simulation and calculation.

To identify differences in the electrothermal conversion characteristics between silicon and CBG-coated silicon, another experiment was conducted. As illustrated in Figure 6c, the temperature and electrical conductivity curves of the two materials did not coincide. When only silicon was used, the curve was drawn from the equation [11]. Below 400 °C, the electrical conductivity of silicon was very low, almost close to 0. But once the temperature exceeded 400 °C, the electrical conductivity of silicon rose rapidly. By comparing with the curve of CBG coated silicon, it was found that the electrical conductivity value of the one with CBG coating was already more than 0 from the beginning. Furthermore, its electrical conductivity began to increase rapidly at 200 °C, which was 200 °C earlier than it was for the condition of silicon. Additionally, the rate of increase in electrical conductivity with temperature was significantly faster for silicon with CBG coating than for silicon. This observation suggests that certain substances may have the potential to modify the electrical conductivity properties of silicon.

According to the experimental model, the composition of the system comprised CBG-coated silicon, quartz glass, and braided copper strips. Note that the CBG-coated silicon was composed of intrinsic silicon and CBG coating, both of which are semiconductor materials; quartz glass, serving as an insulator, and braided copper strips, acting as electrodes, had no effect on the electrical conductivity of silicon. Therefore, the increased electrical conductivity of silicon below approximately 400 °C could be attributed to the CBG coating on its surface. The formation of covalent bonds may lead to the introduction of defects, such as sp_3_ hybridization and distortions, in the graphene crystal, which can impact its thermal and electrical properties. Graphene is recognized as the most conductive material at room temperature, with a sheet resistance of 31 Ω/sq [15] and a conductivity of 10^6^ S/m. CBG coating has similar properties to graphene, so it could combine with silicon efficiently to act as a “thin film resistance heater” for rapid heating. At room temperature, the intrinsic silicon has a very low electrical conductivity of approximately 4 × 10^−4^ (Ω∙m)^−1^, due to few electrons being excited to cross the forbidden energy gap. By contrast, the energy band gap of the CBG coating is 60.2 meV [10], nearly 20 times smaller than that of intrinsic silicon, which means that less energy is required to shift an electron from the valence band to the conduction band. Consequently, the CBG-coated silicon exhibited higher electrical conductivity than the silicon at room temperature. A faster increase in electrical conductivity was observed at lower temperatures, with CBG coating playing a dominant role. As the temperature exceeded 400 °C, the electrical conductivity sharply rose. At this point, the characteristics of silicon were found to be similar to this situation. This is because at low temperatures, the carbon ions in graphene possess low energy. However, as temperature increases, the ions gain energy and begin to oscillate around their mean positions. The oscillating ions collide with the moving electrons more easily, reducing conductivity. The extrinsic region transitions to the intrinsic region, supplying more thermal energy to the intrinsic silicon and CBG coating. This process leads to the breaking of more covalent bonds and the subsequent creation of a larger number of free electrons. As a result of the increased collision activity between free electrons and carbon ions, graphene’s electrical conductivity decreases with temperature [16]. During this period, silicon demonstrated a dominant role. For the speed of electrical conductivity with increased temperature, that of the CBG-coated silicon is still higher than that of silicon. In conclusion, the results indicate that CBG coating played a pivotal role in promoting thermal and electrical conductivity before 400 °C, whereas silicon becomes the dominant factor thereafter.

Additionally, we noticed that the size of the samples affected their temperature and electrical conductivity, as shown in Figure 6d. Sample 1, which follows the same design of silicon with CBG coating (Figure 6c), measured 18 × 18 × 11 mm. On the other hand, Sample 2 was smaller, with dimensions of 18 × 18 × 6 mm. From this comparative analysis, it was observed that the electrical conductivity of Sample 1 escalated more rapidly than that of Sample 2 with an increase in temperature. This suggested that larger sizes lead to quicker increases in conductivity with temperature. Moreover, at equivalent temperatures, the larger samples exhibited enhanced conductivity levels, underscoring the pivotal influence of size on the electrical conductivity behavior of these materials under thermal conditions.

This critical finding enables us to forecast the entire heating process’s temperature and electrical conductivity change trends to facilitate the detection of potential damage to the CBG-coated silicon during the embossing process. If a sudden deviation between the actual and predicted curves could indicate possible damage to the mold core during the hot embossing process, this necessitates its replacement to ensure the fabrication of sustained high-quality products. What is more, it can provide an early prediction of the mold core’s service life. A gradual deviation between the actual and predicted curves of the mold core may suggest that the CBG coating on the silicon mold core is nearing depletion, indicating that the CBG-coated silicon mold core has reached the end of its service life. In this case, removal and recoating of the mold core would be necessary to ensure the continued effectiveness of the non-isothermal glass hot embossing system.

Drawing from the insights gained from our analysis, we sought to further validate above statement through a series of hot embossing experiments. The sample material used for glass hot embossing in this study was K9. The molding process involved four stages: heating, embossing, cooling, and demolding. Initially, an aspheric surface was machined on the external surface of the intrinsic silicon mold core using an ultraprecision single point turning machine (350FG, Moore Nanotech, Swanzey, NH, USA). The parameters of the aspheric surface were determined by the aspheric formula equation.

After machining, the silicon mold core surface was coated with CBG coating using the CVD method, without lubricant and with compressed air cooling. The next step involved the hot embossing of glass, where the silicon mold core and quartz glass were placed in the mold holder and fixed. A round shape glass sample was placed in the pit of the silicon mold core and heated to 700 °C for embossing. The lens forming was carried out using the direct pressing method, and the pressing and demolding processes were completed in a span of 10 s. The upper mold movement was powered by the motor-driven lead screw while the lower mold remained fixed throughout the process.

The glass hot embossing product was analyzed by metallographic microscopy (SOPTOP MX6R). Figure 8a shows the glass product in its normal condition, with a clean surface devoid of air bubbles. Its aspheric profile was in line with that of the silicon mold core, indicating that the surface profile of the mold core has been successfully transferred to the glass product’s surface. However, as the embossing process progresses, stains may start to emerge on the glass surface (Figure 8b). When we compared the electrical conductivity curve with temperature, shown in Figure 8c, we noticed that under abnormal conditions, the temperature for the rapid increase in conductivity was higher than under normal conditions. Subsequently, when comparing the CBG-coated silicon mold core surface, the stains on the glass product surface revealed the peeling of the CBG coating, indicating damage to the CBG coating. This observation was further supported by the alteration in the curve of electrical conductivity changes with temperature under abnormal conditions. If the CBG coating had not been damaged, the curve would align with that of the normal condition. The results reveal the potential for early detection of damage and assessment of the service life of CBG-coated silicon.

## 4. Conclusions

In this study, we investigated the changes in electrical conductivity of CBG-coated silicon with temperature during a non-isothermal glass hot embossing process. By comparing simulation and experimental data, we observed a strong correlation between temperature-dependent variations in electrical conductivity. These findings provide valuable insights into the electrothermal conversion properties of CBG coatings.

(1) At lower temperatures, the electrical conductivity of CBG-coated silicon gradually increases. However, as the temperature rises, there is a sharp increase in electrical conductivity. The behavior of electrical conductivity with temperature is as follows: below 400 °C, the CBG coating is the primary conductor, while above 400 °C, silicon becomes the predominant conducting material.

(2) The variations in temperature and electrical conductivity curves are associated with size changes, indicating that larger volumes exhibit a faster increase in conductivity with increased temperature and achieve higher conductivity levels at equivalent temperatures.

(3) The reduction in CBG coating on the silicon will affect the variation in electrical conductivity with temperature, enabling the detection of damage and prediction of service life through comparison with the normal situation.

The rapid increase in the electrical conductivity of CBG-coated silicon with increased temperature poses challenges in maintaining stable temperature control with constant power input during the non-isothermal hot embossing process. Dynamic changes in current flow can result in uneven temperature conduction and thermal stress in the molding process, leading to distorted aspheric glass lenses and reduced product precision. The predicted trends in temperature and electrical conductivity changes during the heating process can facilitate early detection of potential damage to the CBG-coated silicon and provide an early indication of the mold core’s service life, enabling timely removal and recoating.

## Figures and Tables

**Figure 1 micromachines-15-00673-f001:**
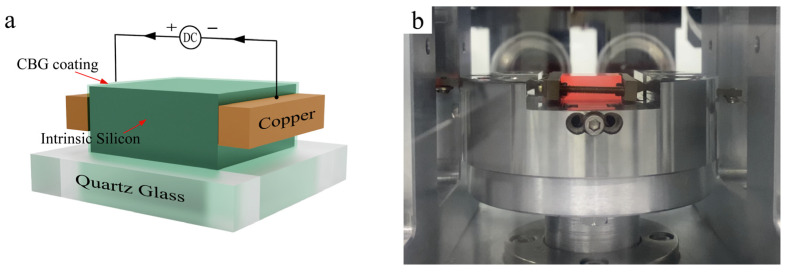
(**a**) Electrical model during the heating process. (**b**) A CBG-coated silicon mold core inside the vacuum chamber during the heating process.

**Figure 2 micromachines-15-00673-f002:**
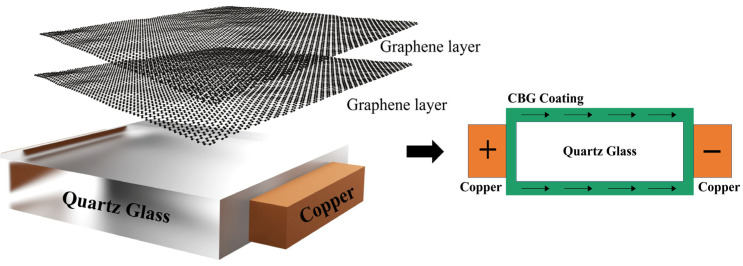
The method of measuring CBG coating resistance.

**Figure 3 micromachines-15-00673-f003:**
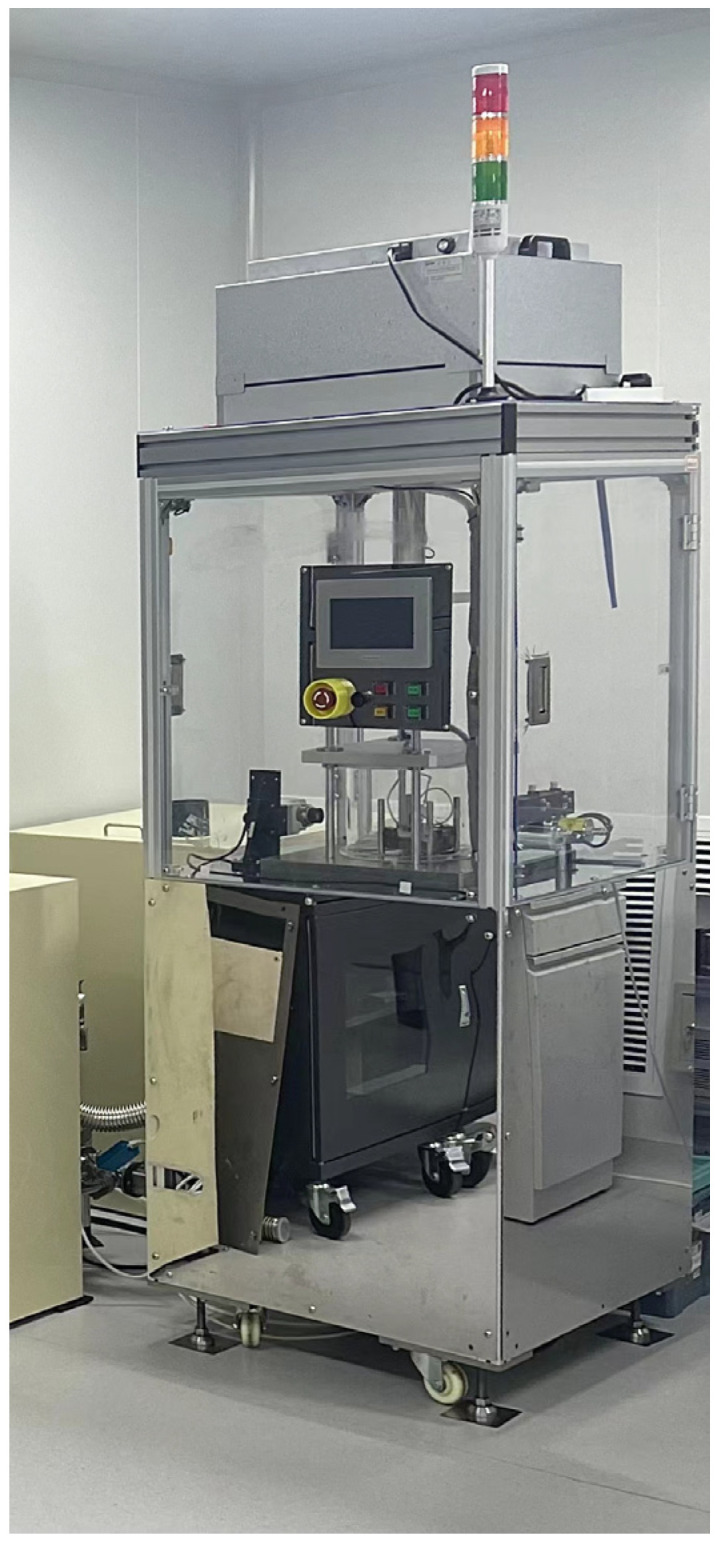
Non-isothermal glass hot embossing system.

**Figure 4 micromachines-15-00673-f004:**
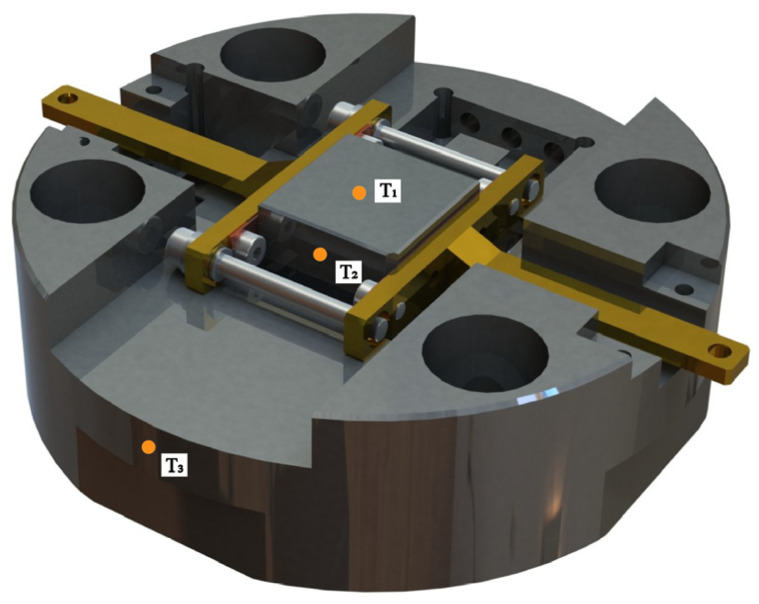
The positions of the T_1_, T_2_ and T_3_ thermocouples.

**Figure 5 micromachines-15-00673-f005:**
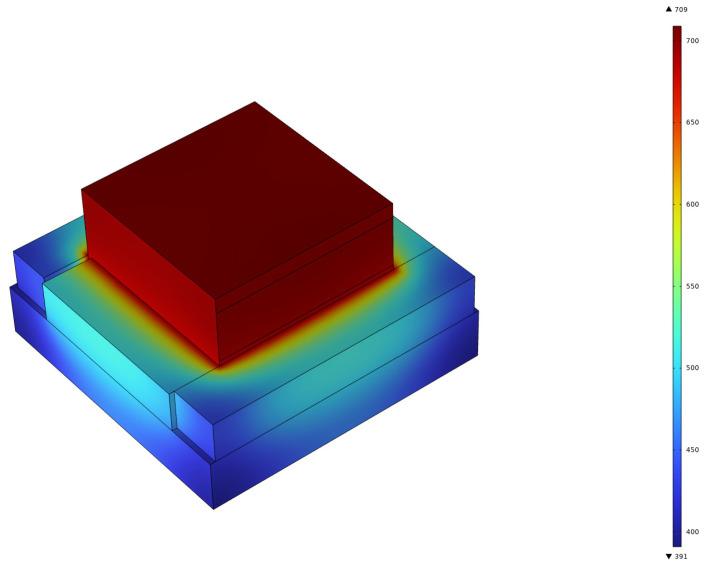
Time = 90 s; the simulation result of the heating process on surface temperature.

**Figure 6 micromachines-15-00673-f006:**
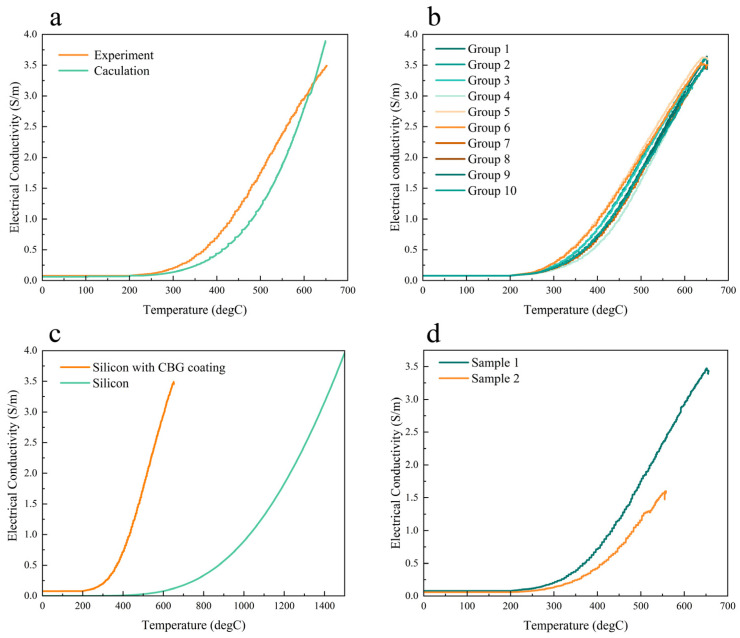
Experiment and calculation results of the temperature and electrical conductivity curve of the CBG coated on a silicon mold. (**a**) Comparison of silicon mold with CBG coating’s temperature-conductivity graph between stimulation and experiment; (**b**) The results of experiments on the influence of existence and different distances of upper mold; (**c**) Comparison of temperature and electrical conductivity curves between silicon and silicon with CBG Coating; (**d**) Comparison of temperature and electrical conductivity curves between different sizes of CBG-coated silicon.

**Figure 7 micromachines-15-00673-f007:**
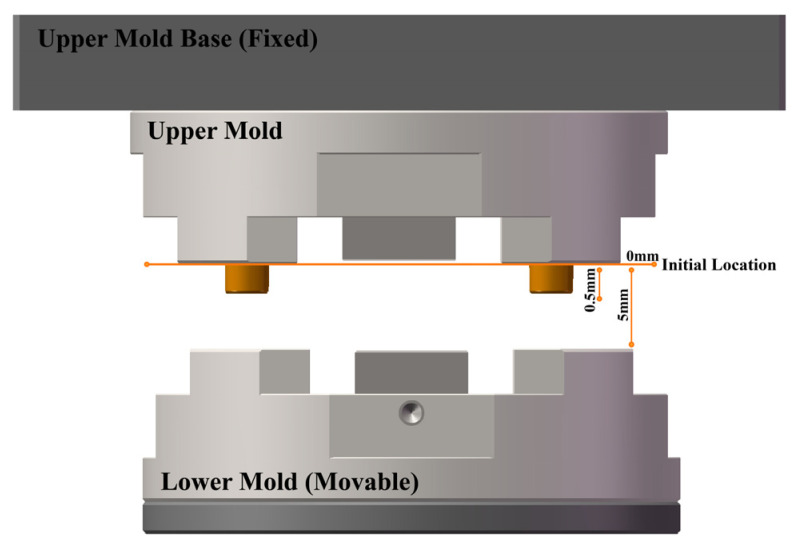
The demonstration of the distance between upper and lower mold in the experiment.

**Figure 8 micromachines-15-00673-f008:**
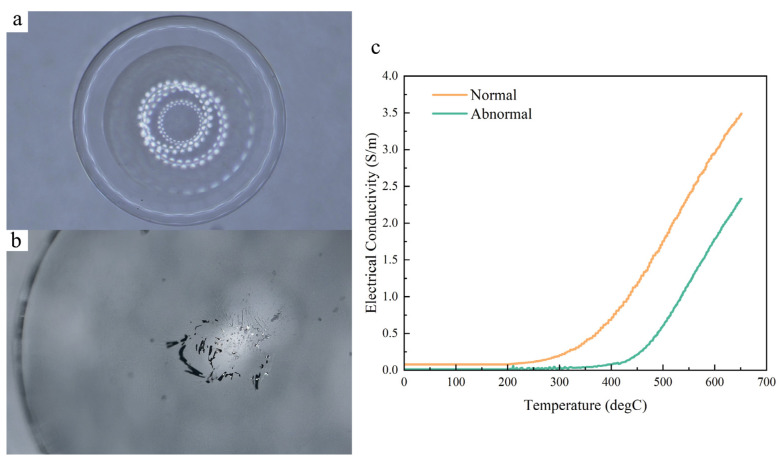
(**a**) Normal aspherical lens sample produced by the glass hot embossing process; (**b**) Abnormal aspherical lens sample produced by the glass hot embossing process, with stains on the surface indicating damage to the mold core; (**c**) Comparison of temperature and electrical conductivity curves between normal (pre-service life) and abnormal (post-service life) conditions.

**Table 1 micromachines-15-00673-t001:** Experiment Design.

Group	Condition	Upper Mold Temperature (°C)	Lower Mold Temperature (°C)	Distance to Initial Location (mm)
1	Only lower mold	\	650	5.0
2	Only lower mold	\	600	5.0
3	Only lower mold	\	650	0.5
4	Only lower mold	\	600	0.5
5	With upper mold	\	650	5.0
6	With upper mold	\	650	0.5
7	With upper mold	650	650	0.5
8	With upper mold	650	650	5.0
9	With upper mold	550	650	0.5
10	With upper mold	550	650	5.0

## Data Availability

The original contributions presented in the study are included in the article, further inquiries can be directed to the corresponding authors.

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
