# Peer review of "Modeling and Experiments on Temperature and Electrical Conductivity Characteristics in High-Temperature Heating of Carbide-Bonded Graphene Coating on Silicon"

_micromachines, 2024, doi:10.3390/mi15060673_

Round 1

Reviewer 1 Report

Comments and Suggestions for Authors

The authors Lihua Li et al. submitted their manuscript titled "Modeling and Experiments on Temperature and Electrical Conductivity Characteristics in High-temperature Heating of Carbide-bonded Graphene Coating on Silicon". In the manuscript, the authors used the finite element method to develop a model for analyzing the relationship between electrical conductivity and temperature, and they aimed to enable high-precision closed-loop temperature control during the glass hot pressing process. The research is practical. However, the research work is not sufficient.

Majors:

1.        More simulation results (for example, at different heating times) should be provided.

2.        In lines 320 to 330, the authors analyze the meaning of the manuscript. In lines 331 to 355, the authors attempted to incorporate hot embossing experiments to enhance the significance of the manuscript. Obviously, there is a significant gap between the content in lines 320 to 330 and lines 331 to 355.

3.        In line 185, the unit should be added to 18*18*11. In line 236, remove “the changing”.

Comments on the Quality of English Language

The quality of English Language is good.

Reviewer 2 Report

Comments and Suggestions for Authors

First of all please add statements proving your research is novell and has an important impact on the area.  Also comment on wheather your study is of an industrial or scientific soundness. To do so explain in the abstract and in the introduction how your study was aimed to improve the state of knowledge. Then in the conclusions please confirm that your predictions were correct and the results you present are of potential interest in the field.

Then I have doubts regarding lines 310-330: the size of the samples surely affects the amount of heat the sample can accomodate, the temperature variation speed etc. please comment on the subject. 

Comments on the Quality of English Language

Please correct some minor yet annoying grammar mistakes. 

Round 2

Reviewer 1 Report

Comments and Suggestions for Authors

The manuscript can be accepted in its current version.

Reviewer 2 Report

Comments and Suggestions for Authors

Dear authors. Eq. 2 obviously can not fot your data. You explain the reasons in the text which os ok. Can you make some fits to Your data end find an equation (a new term shaoul be added probably - some multiplication dependant on T from some point)? That could explain if the temparature from which eq. 2 deviates from experimantal data has something to do with bonds formation, hybridization, ... Also that can help with setting "stable" conditions for industrial applications. 

Round 3

Reviewer 2 Report

Comments and Suggestions for Authors

Accept in present form.